

# A new method for estimating aerosol mass flux in the urban surface layer by LAS

R.M. Yuan[1], T. Luo[1], J. N. Sun[2], H. Liu[1], Y.F. Fu[1], and Z. Wang[3]

[1]School of Earth and Space Sciences, University of Science and Technology of China, Anhui, 230026, China

[2]School of Atmospheric Sciences, Nanjing University, Jiangsu, 210023, China

[3]Deparmment of Atmospheric Science, University of Wyoming, Laramie, WY, 82070, USA

*Correspondence to*: R. M. Yuan (rmyuan@ustc.edu.cn) and T. Luo (luotao@ustc.edu.cn)

**Abstract:** Atmospheric aerosol has a great influence on human health and the natural environment as well as the weather and climate system. Therefore, atmospheric aerosol has attracted significant attention from the society as a whole. Despite consistent research efforts, there are still uncertainties in our understanding of its effects due to poor knowledge of aerosol vertical transport caused by our limited measurement capability of aerosol mass vertical transport flux. In this paper, a new method for measuring atmospheric aerosol vertical transport flux is developed based on the similarity theory of surface layer. The theoretical results show that aerosol mass flux can be linked to the real and imaginary parts of the atmospheric equivalent refractive index structure parameter (AERISP), and the ratio of aerosol mass concentration to the imaginary part of the atmospheric equivalent refractive index (AERI). The real and imaginary parts of AERISP can be measured based on the light propagation theory. The ratio of aerosol mass concentration to the imaginary part of AERI can be measured based on the measurements of aerosol mass concentration and visibility. The observational results show that aerosol vertical transport flux varies diurnally and is related to the aerosol spatial distribution. The maximum aerosol flux during the experimental period in Hefei City was 0.017 mgm$^{-2}$s$^{-1}$, and the mean value was 0.004 mgm$^{-2}$s$^{-1}$. The new method offers an effective way to study aerosol vertical transport over complex environments.

## 1   Introduction

The impacts of atmospheric aerosols on climate change have drawn significant attention from society as a whole (IPCC, 2014). To better understand the aerosols, there are many conventional and routine measurements, such as the measurement of the concentration of aerosol particles for environmental protection (Chang and Lee, 2007;Cruz et al., 2015), the observation networks of ground-based and remote sensing for tropospheric aerosol properties and radiative forcing (to measure the optical





depth, concentration, and physical and chemical properties of aerosol) (Cruz et al., 2015;Dubovik et al., 2002), and some special scientific experiments (Li et al., 2015;Wood et al., 2013). In the past twenty years, great progress has been made in the measurement of concentration, size distribution, and physical and chemical properties of aerosols. However, there are still large uncertainties in quantifying the effects

of atmospheric aerosols on Earth's energy budget by scattering and absorbing radiation and by modifying the amounts and microphysical and radiative properties of clouds (Myhre et al., 2009;IPCC, 2007, 2014). Therefore, more representative and accurate data are required (Chin and Coauthors, 2009;IPCC, 2014). The climatic effect of aerosol was also extensively studied through numerical model simulations, which also must be verified by further directly measured aerosol data, especially aerosol emissions from the

surface (Myhre et al., 2009). The forecast of urban environmental pollution also must directly measure the aerosol source emission (Wu et al., 2012).

So far, we can provide accurate physical and chemical aerosol properties, such as concentration, shape, size, optical properties and chemical components, especially with in situ sampling instruments (Nakayama et al., 2014;Wang et al., 2015). However, other key aerosol processes, such as emission

intensity and vertical transport, which are required for simulations of large-scale atmospheric chemistry transport and forecasting local and regional air quality, are poorly measured. Studies (Seinfeld and Pandis, 2006;Bond et al., 2004) have shown that model simulations of aerosol impacts on the environment and climate require data of aerosol source emission and transport, especially the emission of anthropogenic aerosols, arising primarily from a variety of combustion sources (Li et al., 2009).

There are a few methods to provide aerosol emission data. One method is to perform statistical analyses of aerosol parameters and estimate aerosol flux based on data from sources such as the statistical yearbooks by governments, including activities of power generation (Kirill et al., 2006;Chin and Coauthors, 2009;Zhao et al., 2012). Another method is to estimate aerosol flux based on the method of estimating gas flux (Stull, 1988), such as the Bowen ratio method (Lighthart and Shaffer, 1994). However,

there are still large uncertainties in the upward aerosol flux transported from the ground estimated by the two methods (Bond et al., 2004;Kanakidou et al., 2005).

In recent years, with the wide use of instruments for measuring aerosol particle number concentrations (such as the GP-WCPC3787 particle counter produced by TSI), it is possible to measure the vertical transport flux of aerosol particle number density by the eddy-covariance (EC) method. The

vertical transport flux of aerosol particle number density $F_p$ is expressed as cross-correlation between



vertical wind velocity *w'* and aerosol particle number density *N'* (Ripamonti et al., 2013). Fluctuations of vertical velocity and aerosol particle number density must be measured simultaneously at high temporal resolutions to provide aerosol particle number flux.

Measurements of aerosol vertical transport flux using the EC method have been carried out in many cities in recent years, such as Stockholm (Vogt et al., 2011b), Helsinki (Ripamonti et al., 2013), Lecce (Samain et al., 2012), Munster (Pauwels et al., 2008), and London (Harrison et al., 2012). The EC method has also been used for aerosol particle number concentration flux in the other sites, such as sea-salt aerosol concentration flux measurements in Northern Europe (Brooks and Coauthors, 2009;Sproson et al., 2013).

Such measurement capability has provided new insights regarding atmospheric aerosols, such as a strong correlation between atmospheric aerosol particle flux and traffic flow rate in urban areas (Jarvi et al., 2009), characteristics of sea-salt aerosol transportation, and physical-chemical properties of aerosols (Nemitz et al., 2009). The measurements are mostly taken in cities, which have key anthropogenic sources. Urban measurements are easy to obtain with high reliability and can be used as routine model inputs. Although urban aerosol particle number flux has been measured with the EC method, the results represent aerosol particle number flux only at a single point. However, the underlying surface in urban areas is very complex, with large horizontal inhomogeneities, and single-point measurements are not very representative. Therefore, the development of a new measurement system to provide aerosol flux representing a larger spatial region is of great importance. Furthermore, the parameter measured by the EC theory is the aerosol particle number flux, which is often dominated by a high concentration of smaller particles. However, for many applications, the aerosol mass flux is more important.

The sensible heat flux measurements with the EC method are mature and widely used (Lee, 2004). But also the sensible heat flux can be measured by a general Large Aperture Scintillometer (gLAS) based on the similarity theory and the light propagation theory (Zeweldi et al., 2010). Inspired by that, we explored the potential to measure aerosol flux by using the similarity theory and the light propagation theory.

We recently measured the imaginary part of the atmospheric equivalent refractive index structure parameter (AERISP) based on the light propagation theory (Yuan et al., 2015). The results showed that the imaginary parts of AERISP relate to the turbulent transport process and the spatial distribution characteristics of aerosols. Atmospheric equivalent refractive index (AERI) depends on scattering and





absorption of aerosol particles (Barrera et al., 2007;Calhoun et al., 2010;van de Hulst, 1957), and should have something to do with the mass concentration of aerosol particles. Therefore, similar to the fact that the temperature structure parameter can reflect the sensible heat flux, the imaginary part of AERISP can reflect the aerosol mass flux. This paper will present a new method based on this consideration and present the results from field measurements.

Section 2 provides the theory and method of aerosol transport flux measurement. The experiment is introduced in Section 3. Section 4 presents the field observational results, and a conclusion and discussion are presented in the final section.

## 2 Theory and method

### 2.1 Theory of aerosol mass flux measurement

Experiments showed that small aerosol particles follow the same laws of turbulent motion as air molecules, that is, the fluctuation of particle concentration follows the '-5/3' power law under unstable atmospheric stratification, and the concentration-velocity co-spectra for particle number flux follows the '-4/3' power law (Martensson et al., 2006;Vogt et al., 2011a;Kaimal et al., 1972). Therefore, the distribution of small particles can be regarded as a passive conservative quantity, the same as the temperature field. Then at the separation $r$ of the order of inertial subrange scales, the aerosol mass concentration (denoted as $M_a$) structure function $D_M(r)$ in a locally isotropic field follows the "2/3 law" (Wyngaard, 2010) and has the expression as,

$$D_M(\mathbf{r}) = \overline{[M_a(\mathbf{x}) - M_a(\mathbf{x}+\mathbf{r})]^2} = C_M^2 r^{2/3} \qquad (1)$$

where $\mathbf{r}$ is the separation vector, $\mathbf{x}$ is the position vector, $C_M^2$ is the aerosol mass concentration structure parameter, and the overbar in Eq. (1) indicates spatial average. Under the condition of free convection, the mass concentration in the surface layer will follow the similarity theory (Obukhov, 1971;Wyngaard et al., 1971):

$$\frac{C_M^2 z^{2/3}}{M_*^2} = \frac{4}{3}(-\frac{z}{L})^{-2/3} \qquad (2)$$

where $z$ is the effective height above the reference plane (Defined in Section 2.2), $L$ is the Monin-Obukhov (M-O) length, and defined as $L = \frac{\overline{T}u_*^2}{\kappa g T_*} = -\frac{\overline{T}u_*^3}{\kappa g \overline{\theta' w'}}$ (Stull, 1988), $\overline{T}$ is the average


temperature, $u_*$ is the friction velocity, $T_*$ is the surface-layer temperature scale, $\kappa$ is the Karman constant, which is equal to 0.4, and g is the gravity acceleration. $M_*$ can be regarded as atmospheric aerosol mass concentration scale in the surface layer, which is similar to the surface-layer temperature scale. Equation (2) has the same form as the similarity theory (Obukhov, 1971;Wyngaard et al., 1971)

for temperature structure parameter under the condition of free convection in the surface layer

$\dfrac{C_T^2 z^{2/3}}{T_*^2} = \dfrac{4}{3}(-\dfrac{z}{L})^{-2/3}$, in which $C_T^2$ is the temperature structure parameter. Based on the similarity theory

(Wyngaard et al., 1971) and the definition of M-O length, we can obtain

$$u_*^2 M_*^2 = \frac{3}{4} C_M^2 z^{2/3} (\frac{\kappa g Q z}{\overline{T}})^{2/3} \tag{3}$$

where kinematic heat flux $Q$ can be expressed as the product of friction velocity and temperature scale

by $Q = \overline{\theta' w'} = -u_* T_*$ (Stull, 1988). We can obtain a further expression of aerosol flux from Eq. (3):

$$u_*^2 M_*^2 = (\frac{3}{4})^{3/2} \kappa (\frac{g}{\overline{T}}) C_M^2 (C_T^2)^{1/2} z^2 \tag{4}$$

Similar to the expression of heat flux, the aerosol mass flux can be expressed as $F_a = -u_* M_*$, and

the expression of aerosol mass flux is obtained:

$$F_a = (\frac{3}{4})^{3/4} \kappa^{1/2} (\frac{g}{\overline{T}})^{1/2} (C_M^2)^{1/2} (C_T^2)^{1/4} z \tag{5}$$

In Eq. (5), $C_M^2$ is still unknown. How to obtain this parameter is discussed in the following.

The gases and aerosol particles in the atmosphere can be regarded as a whole as an equivalent medium, and its equivalent refractive index is called atmospheric equivalent refractive index (AERI). AERI consists of the real part $n_{Re}$ and the imaginary part $n_{Im}$. For the atmosphere transparent band, $n_{Re}$ mainly depends on atmospheric temperature, and $n_{Im}$ on aerosol extinction (Details are given in

the Appendix). For a given wavelength (usually constant in an experiment), $n_{Im}$ is related to the component, concentration and size distribution of aerosol particles. From the results of numerical calculation (Jennings et al., 1978;Jennings et al., 1979), even if the concentration of aerosols is constant, the aerosol extinction changes with the refractive index and size distribution of aerosols. Therefore we can conclude that the AERI changes with the refractive index and size distribution of aerosols even if the

concentration of aerosols is constant. In other words, a simple relationship, or a one-to-one corresponding



relationship, does not exist between $n_{\mathrm{Im}}$ and the mass concentration of aerosols. However, experiment

results (Cachorro and Tanre, 1997) show that $n_{\mathrm{Im}}$ (or $\sigma_e$ the aerosol extinction coefficient) and $M_a$

have a good linear relationship (see Sec. 4.2). We can define a parameter as the ratio of aerosol mass

concentration $M_a$ to the imaginary part of AERI $n_{\mathrm{Im}}$,

$$R_{MN} = \frac{M_a}{n_{\mathrm{Im}}} \tag{6}$$

Theory analysis has shown that $R_{MN}$ is related to aerosol particle refractive index, mass density of

the aerosol particles, and particle size distribution. For near-surface aerosols at a given location, we can

treat $R_{MN}$ as a constant owing to the relatively small variations in particle size and aerosol refractive

index (Dubovik et al., 2002). Then there is simply linear between $C_M^2$ with $C_{n,\mathrm{Im}}^2$. Based on Eq. (6),

the relationship $C_M^2 = R_{MN}^2 C_{n,\mathrm{Im}}^2$ can be deduced, and the imaginary part of AERISP $C_{n,\mathrm{Im}}^2$ is defined

as $D_{n,\mathrm{Im}}(\boldsymbol{r}) = \overline{[n_{\mathrm{Im}}(\boldsymbol{x}) - n_{\mathrm{Im}}(\boldsymbol{x}+\boldsymbol{r})]^2} = C_{n,\mathrm{Im}}^2 r^{2/3}$ (Yuan et al., 2015). Thus, Eq. (5) can be written as

$$F_a = (\frac{3}{4})^{3/4} \kappa^{1/2} R_{MN} (\frac{g}{\overline{T}})^{1/2} (C_{n,Im}^2)^{1/2} (C_T^2)^{1/4} z \tag{7}$$

Equation (7) is the theoretical basis of aerosol mass flux measurements. The new method for

measuring of $F_a$ will be presented in the next section.

## 2.2 Measurement method

The friction velocity $u_*$ is defined as $u_*^2 = \sqrt{(\overline{u'w'})^2 + (\overline{v'w'})^2}$ (Stull, 1988). The cross-

correlations between the turbulent velocity components ($\overline{u'w'}$, $\overline{v'w'}$ and $\overline{u'v'}$) or between the

turbulent velocity components and the temperature fluctuation ($\overline{u'T'}$, $\overline{v'T'}$ and $\overline{w'T'}$) can be

calculated by the data from a 3-D sonic anemometer ($u'$, $v'$, $w'$ and $T'$). Then the Monin-Obukhov

(M-O) length $L$ can be calculated (Stull, 1988; Wyngaard, 2010).

Assuming that the small particles meet the passive conservative conditions and satisfy the similarity

theory under the condition of free convection, we can estimate aerosol flux by Eq. (7), which involves

three parameters, $R_{MN}$, $C_T^2$ and $C_{n,\mathrm{Im}}^2$.

To obtain $R_{MN}$, measurements must be taken for $M_a$ and $n_{\mathrm{Im}}$. $M_a$ is easily available from regular



particle air quality measurements by several standard instruments (Gebicki and Szymanska, 2012;Wang

et al., 2012),such as beta ray attenuation. Based on the definition of the aerosol extinction coefficient

$\sigma_e$ (Liou, 2002;van de Hulst, 1957), $n_{\mathrm{Im}}$ can be obtained by using the relationship between $n_{\mathrm{Im}}$ and

the aerosol extinction coefficient $\sigma_e$ (Liou, 2002):

$$n_{\mathrm{Im}} = \frac{\lambda \sigma_e}{4\pi} \qquad (8)$$

where λ is the work wavelength. $\sigma_e$ can be obtained by the visibility measurements (Qiu et al., 2004),

$$\sigma_e = \frac{3.912}{L_V}\left(\frac{0.55e-6}{\lambda}\right)^{\alpha} \qquad (9)$$

where $\alpha$ is the Angstram exponent, and is usually set as 1.

Based on the relationship between $\sigma_e$ and $n_{\mathrm{Im}}$ in Eq. (8), we obtain

$$n_{Im} = \frac{0.55e-6}{4\pi} \cdot \frac{3.912}{L_V} \qquad (10)$$

where the unit of visibility $L_V$ is m. Thus, we can obtain $n_{\mathrm{Im}}$ from visibility measurements.

$C_T^2$ and $C_{n,\mathrm{Im}}^2$ can be measured by a specially made Large Aperture Scintillometer (sLAS) (Yuan

et al., 2015). After a spherical wave propagates over a distance in a turbulent atmosphere, the light

intensity in the receiving end will fluctuate. When the attenuation caused by scattering and absorption

along the propagation path is very weak, light intensity fluctuation depends on the fluctuation of the real

part of AERI along the propagation path. When the attenuation caused by scattering and absorption along

the propagation path is relatively strong, the light intensity fluctuation is also related to the fluctuation of

the imaginary part of AERI along the propagation path. With the spectral analysis method, the sLAS light

intensity fluctuations can be separated into the contributions of the real and imaginary parts of AERI.

The contribution of the real part of AERI corresponds to the high frequencies, whereas the contribution

of the imaginary part of AERI corresponds to the low frequencies, suggesting that the variances resulted

from the real and imaginary parts are independent. Therefore, the light intensity variances induced by the

real and imaginary parts can be detected separately at high frequencies and low frequencies from the

sLAS measurements. The real part of AERISP $C_{n,\mathrm{Re}}^2$ can be calculated from the variance of the high-

frequency part, and the imaginary part of AERISP $C_{n,\mathrm{Im}}^2$ can be calculated from the variance of the low-

frequency part. According to the relationship between the temperature and the real part of AERI, $C_T^2$



can be obtained from the real part of AERISP $C_{n,\mathrm{Re}}^2$. Although the gLAS is widely used to measure sensible heat flux in the propagation path, only the high-frequency part of light intensity fluctuation can be detected, and the low-frequency light intensity is often discarded (Solignac et al., 2012). Thus, a specially designed sLAS ( sea details in Section 3) is needed (Yuan et al., 2015).

In Eq. (7), $F_a$ is proportional to the product of the square root of $C_{n,\mathrm{Im}}^2$ and one-quarter of the power of $C_T^2$. That is, the aerosol mass flux is related to both the spatial distribution of aerosol and the strength of turbulence. It can also be seen from Eq. (7) that the aerosol mass flux is related to $R_{MN}$, which is related to aerosol type and size distribution. The aerosol mass flux is also proportional to a height $z$, which is called the effective height above a reference plane (Evans and De Bruin, 2011). But in fact,

$C_{n,\mathrm{Im}}^2$ and $C_T^2$ are both proportional to four-thirds power of the effective height $z$, so the aerosol mass flux does not vary with height. Even so, the effective height $z$ still must be carefully estimated. The method to estimate the effective height $z$ is the same as the method for the effective height $z$ in estimating the sensible heat flux by a gLAS (Evans and De Bruin, 2011).

### 3     Measurement and data processing

Experiments were conducted on two sites, shown as the shadow part and point P in the southern part of Hefei City in Fig. 1 (a). The shadow patch in Fig. 1 (a) is the campus of USTC (the University of Science and Technology of China) for the light propagation experiments, and point P represents the location for determining the aerosol mass concentration $M_a$ and visibility $L_V$.

     Figure 1 (b) displays the measurement site in the USTC campus corresponding to the shadow part

in Fig. 1(a). The measurement site is a typical urban surface in the area. The roads near the campus are often in heavy traffic. One road to the west of the campus is a viaduct, and one road to the north of the campus has eight lanes. The two roads are two arterial highways in Hefei City. There are trees and four-story buildings in most area of the campus, whose mean height is about 15 m, therefore, the plane at the height of 15 m can be considered to be the reference plane.

The sLAS measurements were performed between one building with a height of 55 m (symbol A in Fig. 1 (b)) and another with height 62 m (symbol B in Fig. 1(b)) The distance between the two buildings is 960 m. The transmitter of the sLAS was placed at the building A, and the receiver was placed at the building B. The propagation path is along the south-north direction. The experiments were conducted on





the 10$^{th}$ floor of the two buildings at the height 18 m above the reference plane. For a typical gLAS measurements, the measurement height is a very important physical quantity and should be carefully measured and calibrated (Evans and De Bruin, 2011). For our measurement of aerosol mass flux, the height is also a very important parameter and can be calculated as 18.0 m. The signal measured by a

gLAS has a relatively large weight in the middle part of the propagating path (Wang et al., 1978). The sonic anemometer measurements showed that the turbulence characteristics over the campus do not exhibit significant inhomogeneity. The measurement height of 18 m above the reference plane is high enough to meet the isotropy assumption (Martensson et al., 2006).

The sLAS was built at USTC based on the instrument concept initially developed by Wang et al.

(1978), and the light wavelength was 0.625 μm. The sLAS is very similar to a gLAS for measuring surface-layer sensible heat flux (Moene et al., 2009;Kleissl et al., 2008). A gLAS often discards the electronic component with frequency lower than 0.2 Hz (Kipp and Zonen, 2007). The bandwidth of the amplifier of the sLAS receiver ranges from 0.001 to 250 Hz and the output signal is sampled at a frequency of 500Hz. The unprocessed raw data files are saved in 20-min interval. The diameters of

transmitting and receiving apertures of the sLAS are both 0.18 m. The emitted light is converged by a transmit lens, and then propagates over 960 m to the receiver. A photodetector is placed at the focus of the receiving lens, which transfers light intensities to electrical signals. The electrical signals are demodulated and amplified by an amplifier. The sampling frequency is set 500 Hz. More details of the sLAS can be found in the previous paper (Yuan et al., 2015).

A meteorology tower is installed at the roof of a building (symbol C in Fig. 1(b)).The tower is close to the light path and its top is 18 m above the reference plane. A Campbell CSAT3 anemometer (manufactured in Utah USA) was mounted at the top of the tower with the same height as the light path. Three-dimensional velocities and temperature fluctuation were recorded and sampled at 10 Hz and can be processed to provide sensible heat flux, momentum flux, and stability near the surface every 30 min.

These measurement data were used to obtain the dimensionless parameter $z/L$ in the surface layer.

At site P in Fig. 1 (a) approximately 3 km from the USTC experimental site, the mass concentrations and visibility data were measured at a height of 6 m above ground. The aerosol mass concentrations were recorded for $M_a$ in Eq. (6) every hour. The visibility data were recorded every 10 minutes and averaged hourly.

The measurement period was from December 20, 2014 to December 29, 2014, a total of 10 days.





The weather remained sunny during the experiment, and the typical properties of aerosol mass vertical transport may be easy to determine.

## 4    Experimental results

To estimate urban aerosol vertical transport flux, we measured meteorological conditions, aerosol mass concentration, visibility, and the real and imaginary parts of AERISP. Finally, we calculated the aerosol mass flux.

### 4.1    Meteorological conditions

To better analyse the characteristics of aerosol vertical transport flux, we provide meteorological conditions during the experiment in Figs. 2 (a)-(e), including temperature, humidity, wind speed, wind direction and dimensionless parameter $z/L$. Figure 2(a) shows that 90 percent of the wind speed is less than 3 ms$^{-1}$, and the maximum wind speed is 5.7 ms$^{-1}$. As shown in Fig. 2 (b), all types of wind directions are detected, with easterly wind and westerly wind dominating. The statistical characteristics of wind speed and wind direction are near the annual mean distributions and are representative of the region. Figure 2(c) provides the temperature variation with time, showing clear diurnal variations. The highest air temperature is 14.9°C, and the lowest air temperature is 0.76°C. During the experimental period, it was rather warm compared to the local temperature climatology, and industrial production and other daily activities were normal. Figure 2(d) shows the temporal variations in relative humidity. As we know, relative humidity is an important factor controlling aerosol particle growth (Flores et al., 2012;Winkler, 1973). Therefore, to study the transport flux of dry aerosol mass from different surface aerosol sources, only measurements with relative humidity less than 60% are used in the analyses to minimize the influence of the water content in the aerosols. Shown in Fig. 2(d), the relative humidity was less than 60% during most of the experimental time and less than 80% all the time. Figure 2(e) provides the dimensionless parameter $z/L$, which shows that the atmosphere is under unstable stratification during the daytime and is stable at night. The turbulence under the unstable condition in the surface layer contributes significantly to the vertical transport of heat, mass, and water vapour (Stull, 1988).

### 4.2    Ratio of aerosol mass concentration to imaginary part of AERI

$R_{MN}$ in Eq. (6) is the ratio of the aerosol mass concentration to the imaginary part of AERI.





Theoretical analysis showed that $R_{MN}$ is related to refractive index,mass density of the aerosol particles,

and aerosol particle size distribution. Figure 3 presents the temporal variations of mass concentration $M_a$

and visibility $L_V$ in the surface layer. The maximum of $M_a$ is 712 μgm$^{-3}$, and the mean of $M_a$ is 67 μgm$^{-3}$.

The maximum visibility $L_V$ is 31 km, and the mean is 13 km. The imaginary part of AERI can be

calculated from visibility by Eq. (9) and is presented in Fig. 4 for measurements with relative humidity

less than 60%. The scatter diagram of the imaginary part of AERI $n_{Im}$ and $M_a$ given in Fig. 4 shows that

there is good correlation between them, with a linear correlation coefficient of 0.94. The linear fit given

in Fig. 4 has a slope of 6216 kgm$^{-3}$. Therefore, $R_{MN}$ is set as 6216 kgm$^{-3}$ in this study to estimate the

aerosol vertical transportation flux.

### 4.3    Temperature structure parameter and the imaginary part of AERISP

To calculate aerosol mass flux with Eq. (7), the temperature structure parameter and the imaginary

part of AERISP must be measured. The temperature structure parameter can be obtained by measuring

the real part of AERISP. Figure 5 shows temporal variations of $C_{n,Re}^2$ and $C_{n,Im}^2$. $C_{n,Re}^2$ in Fig. 5(a)

exhibits a general diurnal variation (Stull, 1988), indicating that turbulence is strong during the daytime

and weak during the night. $C_{n,Im}^2$ does not show the typical diurnal variation. The previous study showed

that $C_{n,Im}^2$ is related to both the strength of turbulence and the distribution of pollution (Yuan et al., 2015).

### 4.4    Aerosol mass flux

With the above-measured parameters, the aerosol mass flux can be calculated and is presented in

Fig. (6). The aerosol mass flux exhibits an obvious diurnal variation. The strong aerosol mass upward

transport occurs at noon, while the weak transport occurs during the night. Aerosol upward transport is

also quite strong during the local traffic rush hours every day (6:00-9:00 and 17:00-19:00), especially

during the mornings of December 22 and 29, 2014, suggesting a large number of vehicles on Monday

mornings. During the experimental period, the mean value is 0.004 mgm$^{-2}$s$^{-1}$ with a maximum aerosol

mass flux of 0.017 mgm$^{-2}$s$^{-1}$.

Although there are no direct measurements for aerosol mass flux for comparison, the aerosol mass

flux measurements can be compared to the aerosol particle number flux by the EC method with a few



assumptions. Jarvi et al. (2009) measured aerosol particle number flux from July 2007 to July 2008 near a road in Helsinki, Finland. The main pollution source of Helsinki is vehicle emissions, which is similar to the main aerosol source in Hefei City. According to Fig. 5 in their paper, the maximum aerosol particle number flux is approximately $663\times10^6$ m$^{-2}$s$^{-1}$ and occurs at noon within urban areas during the winter

season as well. It is reasonable to assume that urban aerosol mainly comprises fine particles and the aerosol particle volume distribution can be described by lognormal distributions with median diameter 0.3 μm and geometric standard deviation 1.7 (Dubovik et al., 2002). Furthermore, assuming the density of aerosol particles $1.46\times10^3$ kgm$^{-3}$(Yin et al., 2015), Jarvi et al. observed a maximum aerosol particle number flux corresponding to the maximum aerosol mass flux of approximately 0.0092 mg m$^{-2}$s$^{-1}$.

Although this is approximately half the maximum mass flux observed in Hefei City, the two measurements are comparable considering that Hefei City is much more polluted than Helsinki most of the time. Although the comparison is not side by side, it indicates that the new method presented here is reasonable.

**5    Conclusion and discussion**

Based on the similarity theory and the light propagation theory, a new method is developed for quantitatively estimating the aerosol mass flux in the urban surface layer. The similarity theory and the light propagation theory can be applied to aerosol transport in the surface layer, which requires that the aerosol particles are small enough to follow the movement of air and that the size distribution of aerosol

particles remains unchanged. For aerosols with residence life in the boundary layer longer than one hour, this requirement is well satisfied. Although there are large coarse particles in the source regions (Dubovik et al., 2002), they typically fall out quickly.

The estimations of the aerosol mass flux by the proposed method require measurements of the following parameters: temperature structure parameter, imaginary part of AERISP and the ratio of aerosol

mass concentration to the imaginary part of AERI, which can be measured experimentally. According to the 10-day measurements, the aerosol mass vertical transport flux shows the typical diurnal variation. During the experimental period, it was mainly sunny. Thus, the convective turbulence greatly contributed to aerosol mass flux and resulted in large aerosol mass flux during the daytime. Moreover, the aerosol mass flux was rather high during local traffic rush hours. The measurement site is in Hefei City. This city



has a population of about 3,500,000 and more than 600,000 vehicles. Thus, vehicle emissions provide one of the main pollution sources. Our results indicated that the vertical transport flux of aerosol mass is controlled by both turbulence transport and aerosol emissions, which is expected physically.

Although there are no other direct measurements available for evaluation, the measurements were

indirectly compared to aerosol particle number flux measured by the EC method at another location, which revealed a comparable maximum mass flux. In the near future, we will explore the opportunity for side-by-side comparison. Compared to the EC method, measuring aerosol mass flux by the light propagation method has some advantages, such as no need for building a high tower or large spatial coverage.

The error of aerosol mass flux measurements results from theory and instrument. Similar to the gLAS for measuring sensible heat flux at the surface layer, instrument measurement error may be attributed to electronic or optical problems, which, for example, include a poor focal alignment of the receiver detector and the transmitter diode, calibration of electronics, and effective diameter of the gLAS transmitter and receiver (Moene et al., 2009;Kleissl et al., 2009;Kleissl et al., 2008). These errors can be

minimized through careful experiment setup and data quality control. Theory error includes surface-layer similarity theory application, difference between gas molecular and large aerosol particles, and deviation from linear relationship between $n_{\mathrm{Im}}$ and $M_a$. Under neutral and stable conditions, the aerosol mass flux is difficult to assess (Pauwels et al., 2008;Samain et al., 2012). In fact, the aerosol vertical transport flux under the stable condition is very weak. Measurement results were accordance with the fact. Thus,

lacking accurate aerosol vertical transport flux under the stable condition will not introduce a significant error in overall aerosol mass flux estimations. The ratio of aerosol mass concentration to the imaginary part of AERI is assumed to be constant, which is a reasonably good assumption for a given location with a dominant aerosol type, such as urban aerosols. However, the variations in the ratio $R_{MN}$ will introduce errors in the aerosol mass flux measurements. The ratio $R_{MN}$ depends on the refractive index and size

distribution of aerosol particles. Therefore, the ratio can be determined locally when the approach is applied to different locations. Of course, if there exists some variation in aerosol particle refractive index and particle size distribution, $R_{MN}$ can be obtained by measurement simultaneously with the imaginary part of AERISP $C_{n,\mathrm{Im}}^2$, so that real-time $R_{MN}$ can be obtained. The large aerosol particles cannot follow





the movement of air well (Seinfeld and Pandis, 2006;Vogt et al., 2011a), which is also one of error sources for aerosol mass flux measurements. However, these large aerosols fall out very quickly and have a small impact on estimated aerosols, which can stay in the atmosphere over a day. All these error estimations should be discussed quantitatively in the future.

**Acknowledgements**

This study was funded by the National Natural Science Foundation of China (41475012) and partially by the Jiangsu Provincial Collaborative Innovation Centre of Climate Change. We also thank two anonymous reviewers for their constructive and helpful comments.

**Appendix: Atmospheric equivalent refractive index (AERI) and aerosol mass concentration**

When the gases and aerosol particles in the atmosphere are considered as a whole, the AERI $n_{eff}$ can be written as follows (van de Hulst, 1957;Barrera et al., 2007;Calhoun et al., 2010):

$$n_{eff} = n_m + \frac{i}{k} \frac{2\pi}{(n_m k)^2} \int_0^\infty S(0) \frac{dN}{dD} dD \qquad (A1)$$

Equation (A1) includes two parts. The first is the contribution of air molecules, where $n_m$ is the air refractive index. The second represents the scattering and absorption of aerosol particles, where $k$ is the

light wave number ($k=2\pi/\lambda$, where $\lambda$ is the work wavelength) and $i$ is an imaginary number. $S(0)$ is the aerosol forward scattering function (0 in the bracket represents the scattering angle) and can be calculated from the Mie scattering theory. $N$ is the amount of aerosol particles per unit volume, $D$ is the aerosol diameter, and $dN/dD$ is the size distribution function of aerosol particles.

For the atmosphere transparent band, the absorption of gases can be ignored because the extinction

caused by molecular scattering is relatively small compared with that caused by atmospheric aerosols in the urban area. Therefore, there is only the real part of the refractive index of air in Eq. (A1). The real and imaginary parts of AERI ( $n_{Re}$ and $n_{Im}$ , $n_{eff} = n_{Re} + i \cdot n_{Im}$ ) can then be derived as

$$n_{Re} = n_m - \frac{1}{k} \frac{2\pi}{(n_m k)^2} \int_0^\infty \text{Im}[S(0)] \frac{dN}{dD} dD \qquad (A2)$$

$$n_{Im} = \frac{1}{k} \frac{2\pi}{(n_m k)^2} \int_0^\infty \text{Re}[S(0)] \frac{dN}{dD} dD \qquad (A3)$$

As shown in Eqs. (A2)-(A3), $n_{Re}$ is determined by the refractive index of air, the imaginary part



of the forward scattering function of aerosol particles, and the distribution of aerosol particles. $n_{Im}$ is

determined by the real part of the forward scattering function of aerosol particles and the distribution of

aerosol particles. According to the magnitude comparisons and the variation ranges of the two terms in

Eq. (A2), $n_{Re}$ is mainly determined by the refractive index of air (Liou, 2002;Tatarskii, 1961). Thus,

5 the second term on the right-hand side of Eq. (A2) can be ignored. Equation (A3) shows that there is a

relationship between $n_{Im}$ and the aerosol extinction coefficient $\sigma_e$, i.e., $n_{Im} = \lambda \sigma_e / 4\pi$ (Liou, 2002).

The aim of analyzing the imaginary parts of AERI ($n_{Im}$) is to get information of the aerosol mass

concentration. The mass concentration of aerosols, $M_a$, can be expressed as

$$M_a = \rho \frac{1}{6} \pi \int_0^\infty D^3 \frac{dN}{dD} dD \qquad (A4)$$

10 where $\rho$ is the mass density of the aerosol particles.

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



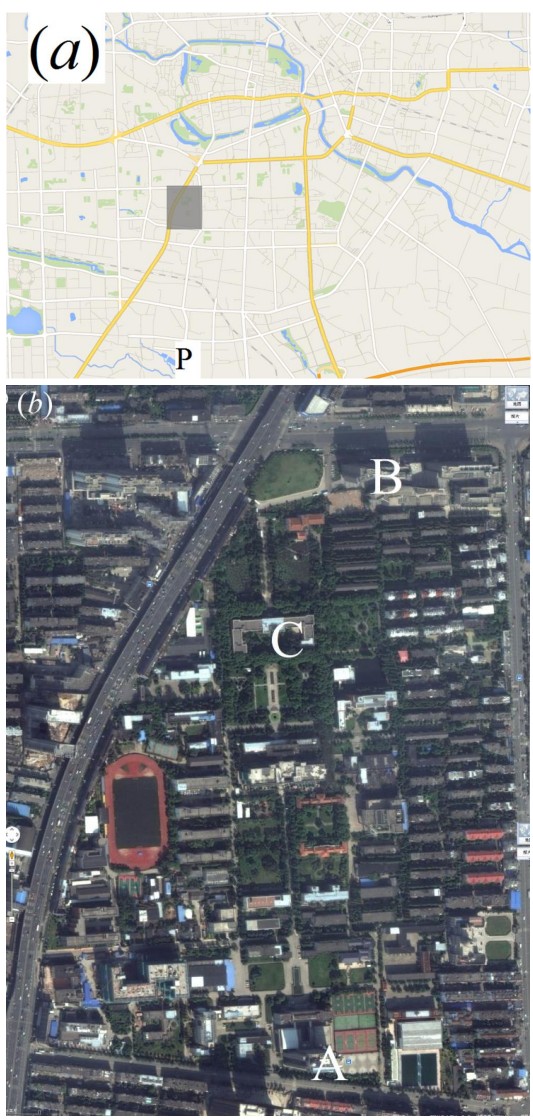

Figure 1. Photographs of the measurement site. (a) Map of Hefei City and (b) expanded view of the measurement site on the USTC campus, which is marked with a rectangle in (a). Point P in (a) indicates the site from which visibility and aerosol mass concentration measurements were obtained. Points A and B in (b) show the locations of the transmitter and receiver, respectively. Point C in (b) marks the

10 meteorological tower position. There are four heavy traffic roads surrounding the measurement site. Figures 1a and b © Google.





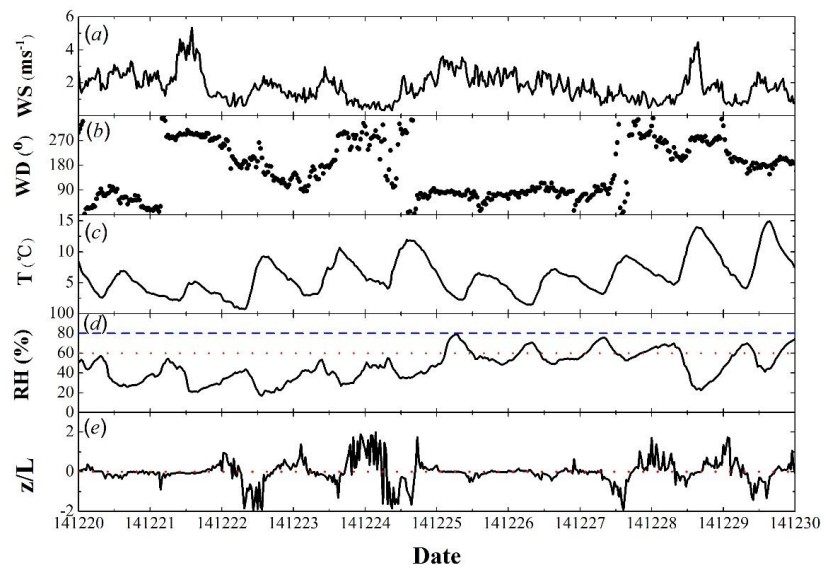

Figure 2. Temporal variations in the (a) wind speed, (b) wind direction, (c) air temperature, (d) relative humidity (RH) and (e) dimensionless parameter ($z/L$) observed from 20-29 December 2014. The dashed line in (c) denotes RH 80%, and the dotted line in (c) denotes RH 60%. The dotted line in (e) denotes $z/L$=0. Details can be found in the text.

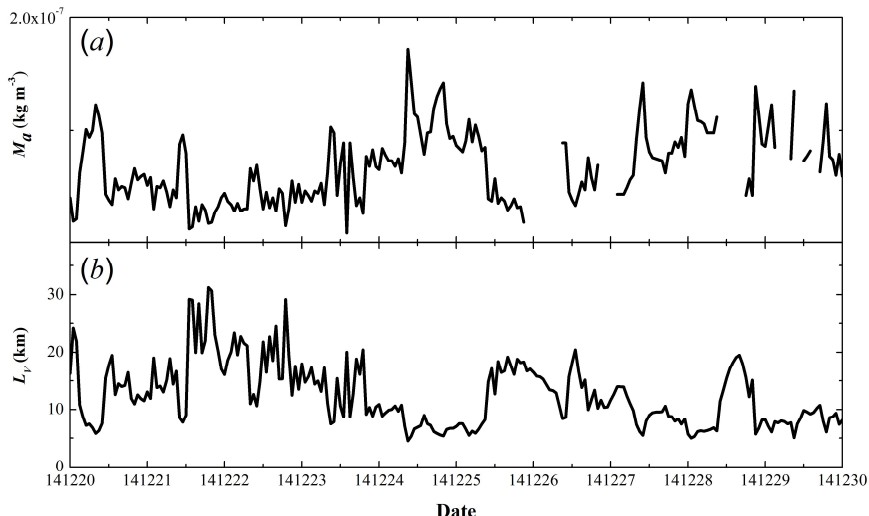

Figure 3 Temporal variations in the aerosol mass concentration $M_a$ (a) and visibility $L_v$ (b) observed from 20-29 December 2014.



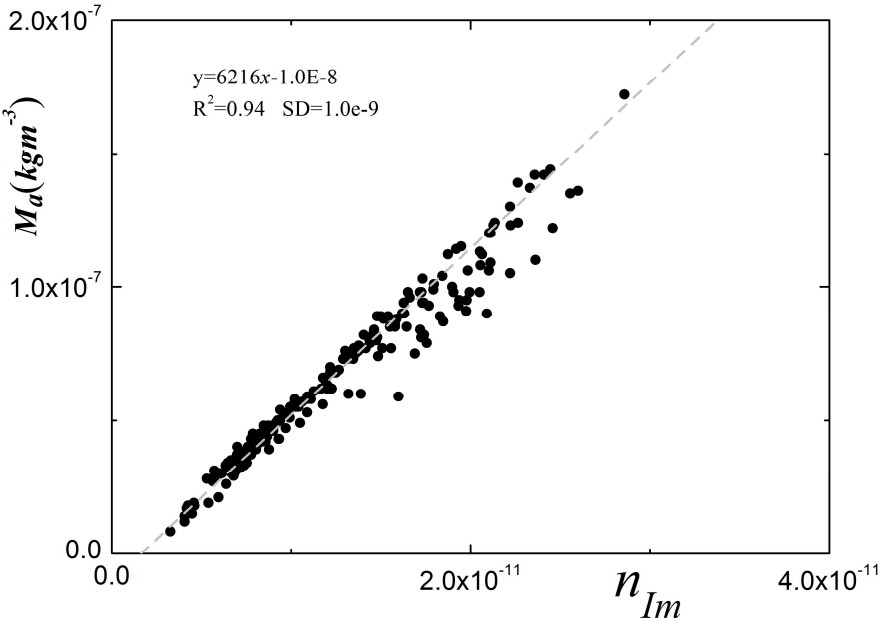

Figure 4 Scatterplots of aerosol mass concentration $M_a$ vs. the imaginary part of AERI calculated from

Fig. 3(b) by $n_{Im} = \dfrac{0.55e-6}{4\pi} \dfrac{3.912}{L_V}$ .

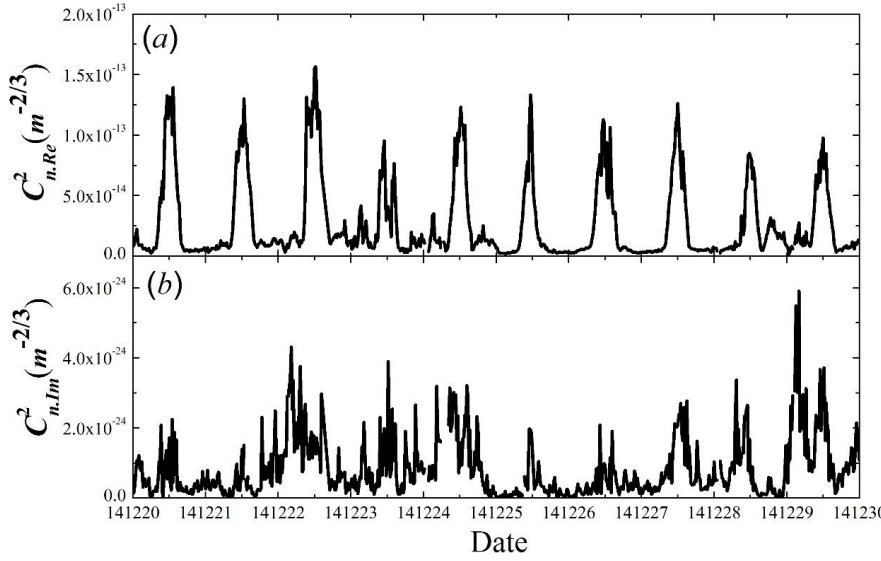

Figure 5. Temporal variations in the real part of the AERISP (*a*), and imaginary part of the AERISP (*b*) observed from 20-29 December 2014.



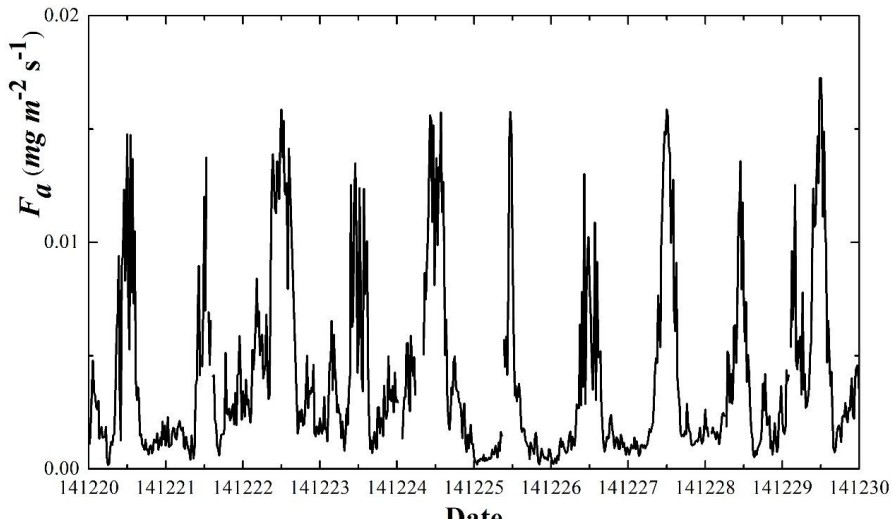

Figure 6. Temporal variations in the mass flux of aerosol observed from 20-29 December 2014.