# Peer review of "A new method for estimating aerosol mass flux in the urban surface layer by LAS"

_Atmospheric Measurement Techniques, 2015_

## Referee Comment (RC1) · Anonymous Referee #1 · 20 Jan 2016

Comments:

The manuscript presents a novel method for estimating aerosol vertical transport flux in the urban surface layer by LAS. This subject is interesting to the air quality modeling and light propagation communities. The theoretical analysis and experimental observation demonstrate the rationale of this new method. However, some technical details are confused.

Technical issues:

1. About the Eq.(8) for the relationship of aerosol absorption and the imaginary part n(Im). The application of this formula in this study is confused because the visibility-derived aerosol extinction can not be related to the aerosol imaginary part n(Im) in this way. In other words, the visibility-derived aerosol extinction is much different from

the aerosol absorption due to the aerosol scattering contribution. In this study, for the visibility-derived aerosol extinction in the open atmosphere, the aerosol scattering coefficient is often dominated in comparison to the aerosol absorption. On the other hand, for the filter-based techniques or integrated plate measured aerosol absorption (Moosmüller, et al., 2009), it can be referred to or equal to the aerosol extinction because the aerosol scattering might be small enough to be ignored.

In addition, the citations or the reference for this formula should be given with the page number.

2. Page 10, Line 10. What is the means of parameter "z/L"? Why it represents the atmospheric stability? How to get it?

3. Is it possible to make the vertical wind velocity measurement in the meteorological tower? It may help explain the vertical transport of aerosol mass.

Minors:

1. Page-12, Line 2-3 "which is similar to the main aerosol source in Hefei City". Please delete it. The air pollution mechanism and process in Hefei might be more complex and much heavier.

2. Some citations or the references from the books should be given with the page number.

References: H. Moosmüller, R.K. Chakrabarty, W.P. Arnott, Aerosol light absorption and its measurement: A review, Journal of Quantitative Spectroscopy and Radiative Transfer, Volume 110, Issue 11, July 2009, Pages 844-878, ISSN 0022-4073, http://dx.doi.org/10.1016/j.jqsrt.2009.02.035. (http://www.sciencedirect.com/science/article/pii/S0022407309000879)

---

## Referee Comment (RC2) · Anonymous Referee #2 · 15 Feb 2016

The manuscript is innovative to define the Atmospheric Effective Refractive Index (AERI) and the Atmospheric Effective Refractive Index Structure Parameter (AERISP), then, use the imaginary part of AERI to derive aerosol mass concentration and, particularly, its vertical flux. This is very important for numerical simulation of air quality and for the regional climate models. The results are generally acceptable.

Revision suggestions:

1. It may not be proper to differentiate the common used LAS and the USTC one by 'gLAS' and 'sLAS'. The so-called sLAS, except a slightly shorter IR wavelength, others (especially hardware) are basically similar. The authors stressed the bandwidth of the amplifier and a higher sampling rate, however, the effects particularly in the low frequency region, which may be important for the detection of the imaginary part

of AERI, are not clear. By the way, from the theory of LAS, the effective eddies are with the size about LAS aperture (here $\sim$0.18 m, which is also in the inertial part of turbulence spectra). The contributions from high frequency smaller eddies and low frequency larger eddies are actually minor. This can be easily checked by the refractive index power spectra.

2. As mentioned in later parts, the observation was conducted in late December. Do you think the using of 'free convection' approximation (Eqs. (3), (4), etc.) is proper? As we know, using 'free convection' in the LAS calculation of sensible heat flux may induce an underestimation of 20-30%. How much the error would be induced in the estimation of aerosol concentration and flux in this method? It would be not so difficult to use directly the general similarity formulas.

3. It would be better to have some method used in Hefei to validate the results obtained with this new method.

4. 'Abstract' line 5-6, '...a new method for measuring atmospheric aerosol vertical transport flux is developed based on the similarity theory of surface layer'. This is incomplete. Actually, this work was also based on, especially, the observations & studies of the effective refractive index (and the theory of light propagation in turbulent atmosphere).

---

## Author Comment (AC2) · 10 Apr 2016

The comment was uploaded in the form of a supplement:
http://www.atmos-meas-tech-discuss.net/amt-2015-301/amt-2015-301-AC2-supplement.pdf

---

## Author Comment (AC1)

Comments:

The manuscript presents a novel method for estimating aerosol vertical transport flux in the urban surface layer by LAS. This subject is interesting to the air quality modeling and light propagation communities. The theoretical analysis and experimental observation demonstrate the rationale of this new method. However, some technical details are confused.

Technical issues:

1. About the Eq.(8) for the relationship of aerosol absorption and the imaginary part n(Im). The application of this formula in this study is confused because the visibility-derived aerosol extinction can not be related to the aerosol imaginary part n(Im) in this way. In other words, the visibility-derived aerosol extinction is much different from the aerosol absorption due to the aerosol scattering contribution. In this study, for the visibility-derived aerosol extinction in the open atmosphere, the aerosol scattering coefficient is often dominated in comparison to the aerosol absorption. On the other hand, for the filter-based techniques or integrated plate measured aerosol absorption (Moos-müller, et al., 2009), it can be referred to or equal to the aerosol extinction because the aerosol scattering might be small enough to be ignored.

In addition, the citations or the reference for this formula should be given with the page number.

Answer: According to the theory of small particle scattering (Liou, 2002), the extinction cross section of one particle is given by (Eq. (5.2.92) in Page 189 in Liou's book),

$$\sigma_e = \frac{4\pi}{k^2} \mathrm{Re}[S(0)] \tag{1}$$

So, for aerosols with a size distribution of $dN/dD$, the total extinction coefficient can be calculated as,

$$\beta_e = \frac{4\pi}{k^2} \int_0^\infty \mathrm{Re}[S(0)] \frac{dN}{dD} \, dD \tag{2}$$

Comparison to Eq. (A3) gives Eq. (8), namely,

$$n_{\mathrm{Im}} = \frac{\lambda \beta_e}{4\pi} \tag{3}$$

It is noted that the variable $n_{im}$ in Eq. (8) links with the extinction, which is the sum of scattering

and absorbing.

Equations (1) and (2) given here have been added in the Appendix as equations (A4) and (A5), and relative statements were added in the text too. Please see Page 17 Line 3-7.

2. Page 10, Line 10. What is the means of parameter "z/L"? Why it represents the atmospheric stability? How to get it?

Answer:

The definition of non-dimensional parameter 'z/L' (here, $z$ is the effective height above the reference plane (=18m in this study); $L$ is the Monin-Obukhov (M-O) scale and defined as

$$L = \frac{\bar{T} u_*^2}{\kappa g T_*} = -\frac{\bar{T} u_*^3}{\kappa g \overline{\theta' w'}})$$ characterizes the turbulence processes in the surface layer (Stull, 1988). The

definition was provided in Section 2.

Please see Page 4 Lines 25-26.

According to Stull (1988), the dimensionless parameter $z/L$ measures the relative contributions

from convection ($\overline{w'\theta'}$) and shear movement ($u_*$) to the turbulence. When z/L <0, it is unstable

stratification ($\overline{w'\theta'}$ >0). When z/L >0, it is stable stratification ($\overline{w'\theta'}$ <0). And when z/L close to 0, it is neutral stratification ($\overline{w'\theta'} \to 0$).

In this study, Campbell CSAT3 3-D anemometer can provide fluctuation data ($u'$, $v'$, $w'$ and $T'$). Then cross-correlations between velocity components ($\overline{u'w'}$, $\overline{v'w'}$ and $\overline{u'v'}$) or between velocity components and temperature fluctuation ($\overline{u'T'}$, $\overline{v'T'}$ and $\overline{w'T'}$) can be calculated. And then the

friction velocity $u_*$ can be calculated based on the expression $u_*^2 = \sqrt{(\overline{u'w'})^2 + (\overline{v'w'})^2}$. So

dimensionless parameter $z/L$ can be obtained based on M-O length definition(Stull, 1988;Wyngaard, 2010).

Related statements are added from Page 8 Line 20 to Page 9 Lines 3-9.

Stull, R. B.: An Introduction to Boundary Layer Meteorology, Reidel Publishing Co., Dordrecht, 666 pp., 1988.

Wyngaard, J. C.: Turbulence in the Atmosphere, Cambridge University Press, New York, 393 pp., 2010.

3. Is it possible to make the vertical wind velocity measurement in the meteorological tower? It may help explain the vertical transport of aerosol mass.

Answer: Vertical wind speed measurements were available from a CAST3 sonic anemometer. The following figure shows the variations of vertical wind speed during Dec. 26, 2014, 11:00-12:00. The vertical wind speeds always fluctuate and the mean value of vertical wind is close to 0 with a 30-min or 1-hour averaging.

[Figure]

Figure 1 The temporal evolution of vertical wind speed during Dec. 26, 2014, 11:00-12:00. Sample frequency is 10Hz. The red line in the figure indicates zero.

The vertical transport flux can be expressed as $\overline{wa}$, where $w$ is the vertical velocity, $a$ is the aerosol mass. By using the Reynolds decomposition ($w = \overline{w} + w', a = \overline{a} + a'$, where the overbar denotes the mean field of the variable, and the prime denotes fluctuating part of the variable), $\overline{wa} = \overline{w} \cdot \overline{a} + \overline{w'a'} \approx \overline{w'a'}$, because $\overline{w}$ is close to 0 m/s. Therefore, the vertical velocity only cannot explain the vertical transport of aerosol mass.

Minors:

1. Page-12, Line 2-3 "which is similar to the main aerosol source in Hefei City". Please delete it. The air pollution mechanism and process in Hefei might be more complex and much heavier.
**Answer**: We deleted "which is similar to the main aerosol source in Hefei City".

2. Some citations or the references from the books should be given with the page number.
**Answer**: We added the page number
References: H. Moosmüller, R.K. Chakrabarty, W.P. Arnott, Aerosol light absorption and its measurement: A review, Journal of Quantitative Spec-troscopy and Radiative Transfer, Volume 110, Issue 11, July 2009, Pages 844-878, ISSN 0022-4073, http://dx.doi.org/10.1016/j.jqsrt.2009.02.035. (http://www.sciencedirect.com/science/article/pii/S0022407309000879
**Answer**: We added the reference.